# Distinct tumor genomic signatures underlie canine macrophage polarization

**Rachel V. Brady** [1¤*], **Sunetra Das** [2], **Dawn L. Duval** [1,2,3], **Kristen B. Farrell** [2], **Eric P. Palmer** [4], **Douglas H. Thamm** [1,2,3]

**1** Cell and Molecular Biology Graduate Program, Colorado State University, Fort Collins, Colorado, United States of America, **2** Department of Clinical Sciences, Flint Animal Cancer Center, College of Veterinary Medicine and Biomedical Sciences, Colorado State University, Fort Collins, Colorado, United States of America, **3** University of Colorado Cancer Center, Anschutz Medical Campus, Aurora, Colorado, United States of America, **4** Department of Microbiology, Immunology and Pathology, Colorado State University, Fort Collins, Colorado, United States of America

¤ Current address: Department of Surgical and Radiological Sciences, School of Veterinary Medicine, University of California-Davis, Davis, California, United States of America

* rbrady@ucdavis.edu

## Abstract

Tumor-associated macrophages (TAMs) drive cancer progression and metastasis. However, the mechanisms by which tumor cells shape TAM phenotypes in canine cancers remain poorly understood. We investigated correlations between cancer cell gene expression and macrophage polarization to identify potential biomarkers and therapeutic targets. Tumor-conditioned media from 25 canine cancer cell lines were applied to monocyte-derived macrophages from three canine donors for 24 hours. Following washout, supernatants were analyzed for immunomodulatory cytokines and chemokines. Each cell line's polarization capacity was ranked using modified z-scores, then correlated with RNA-sequencing data through Spearman's correlation and differential expression analysis. Cancer cell lines showed marked heterogeneity in macrophage polarization capacity, largely independent of histologic type. *MVB12A,* a gene involved in exosome biogenesis, strongly correlated with vascular endothelial growth factor (VEGF) stimulation, suggesting exosome-mediated polarization mechanisms. Exosome fractionation experiments confirmed that purified exosomes induced significantly more macrophage VEGF secretion than other conditions, and high-*MVB12A* cell lines showed greater VEGF enrichment in exosomes. *C-C motif chemokine ligand 3* (*CCL3*) was strongly correlated with tumor necrosis factor-alpha (TNF-α) secretion exclusively in histiocytic sarcoma cells, and recombinant CCL3 induced dose-dependent TNF-α secretion from macrophages. High-polarizing cell lines exhibited upregulation of macrophage activation, epithelial-to-mesenchymal transition (EMT), and metabolic reprogramming genes, and downregulation of immune surveillance and cell adhesion genes. Gene set enrichment analysis confirmed pathways for immune suppression, EMT, and extracellular matrix remodeling. These findings identify exosome-associated VEGF stimulation as a previously uncharacterized mechanism in canine tumors and

**Data availability statement:** The Analysis code and raw data underlying this study are available on GitHub (https://github.com/rbrady783/TCM-macrophages) and archived on Zenodo (https://doi.org/10.5281/zenodo.18499208).

**Funding:** NIH Ruth L. Kirschstein National Research Service Award (NRSA) Institutional Research Training Grant (T32OD010437) awarded through Colorado State University (Rachel V. Brady). University of Colorado Anschutz Medical Campus Cancer Center Genomics Shared Resource Core Facility (Cancer Center Support Grant P30CA046934). A gift from the Anschutz Foundation (Douglas H. Thamm).

**Competing interests:** The authors have declared that no competing interests exist.

highlight CCL3 as a potential histiocytic sarcoma-specific driver of macrophage TNF-α secretion. Further validation in canine clinical cohorts will determine whether these pathways can serve as biomarkers or therapeutic targets in veterinary oncology.

## Introduction

The tumor microenvironment (TME) drives cancer progression via tumor growth, metastasis, and treatment resistance [1]. Many new cancer therapies are TME-centric rather than cancer cell-centric, highlighting the importance of the TME [2]. There is major variability in the TME, both between and within tumor types [3]. One contribution to this variability is the immune cell compartment. Tumor-associated macrophages (TAMs) are the most abundant myeloid cells in the TME and comprise a heterogeneous group with markedly diverse phenotypes [4]. They are being investigated as therapeutic targets and biomarkers in cancer. However, despite major advances in understanding TAM biology in human and murine systems, the mechanisms by which tumor cells shape macrophage phenotypes in canine cancers remain poorly understood [5].

Cancer is the leading cause of death for senior dogs, with an estimated six million dogs diagnosed every year in the United States [6,7]. While dogs and people develop similar cancers at the molecular, cellular and tissue level, dogs also represent a distinct clinical hurdle warranting investigation in their own right [7]. Because their tumors arise spontaneously and grow under authentic immune surveillance, they capture the biological complexities of treatment resistance, metastasis, and host-tumor-immune dynamics [8]. Incorporating canine clinical trials into the drug discovery pathway has many advantages for both species. Several FDA-approved cancer drugs have been researched and approved through the parallel efforts of human and veterinary researchers [9].

Unfortunately, immunotherapy has not yet advanced for canine patients to the same extent as humans [10]. While human checkpoint inhibitors and monoclonal antibodies have transformed the treatment of many cancers in people, their canine counterparts remain limited. Rituximab, the anti-CD20 antibody that improved human lymphoma treatment, fails to bind to canine CD20 due to a single amino acid difference. Canine-specific alternatives remain non-commercialized [7,10,11]. To date, only one checkpoint inhibitor has been conditionally licensed for use in dogs [11]. Chimeric antigen receptor-T cell therapy, now standard of care for some human hematologic malignancies, is in its infancy for canine cancer. It has been hampered by many translational limitations, including significant manufacturing and logistical challenges [10,12]. Several key pieces of basic immunology also remain poorly understood in dogs. For example, canine natural killer (NK) cells are poorly characterized compared to their human counterparts [10]. Biomarkers predicting immunotherapy responses in dogs are essentially nonexistent [11]. These gaps reflect limited investment in canine-specific reagent development, lack of funding for large canine trials, and the broader challenge of translating immunotherapeutic advances across species [8].

Among these understudied areas, TAMs represent a particularly promising yet underexplored therapeutic avenue in dogs. TAMs are usually described as having an "M2-like" or immunosuppressive phenotype, although the dichotomous M1/M2 categorization is overly simplistic. Still, most TAM phenotypes promote tumor progression. They drive angiogenesis, facilitate tumor cell invasion and metastasis, and enhance immune evasion [13]. While there are dozens of TAM-targeted therapies in early phase clinical trials in people, realizing the potential for TAM-targeting in canine cancers is still in its infancy, as many fundamental questions remain poorly understood [5]. For example, it is not known what tumor-intrinsic features predict the capacity to polarize macrophages toward immunosuppressive phenotypes and thereby partially shape the tumor-promoting TME.

In this study, we introduce a novel method for TME biomarker discovery. We aimed to correlate canine macrophage phenotypes after polarization with tumor-conditioned media (TCM) with transcriptomic data from canine cancer cell lines. We hypothesized that canine cancer cell lines would have varying abilities to polarize primary macrophages irrespective of their tumor of origin. We also hypothesized that the strongly polarizing cell lines would have relevant differentially expressed genes (DEGs), useful as potential biomarkers or therapeutic targets.

## Materials and methods

### Cell line maintenance and validation

Cell lines were cultured in DMEM or RPMI (Gibco, Grand Island, NY) supplemented with MEM vitamins (Corning), 2 mM L-glutamine, 1 mM sodium pyruvate, and 10–20% fetal bovine serum (FBS, Peak Serum, Bradenton, FL) (D10/R10/R20). Cultures were maintained at 37 °C with 5% $CO_2$ and routinely passaged. All lines were confirmed to be of canine origin and mycoplasma-free [14]. Cell lines were from the Flint Animal Cancer Center (FACC) Canine Tumour Cell Line Panel and were sourced as previously described [15]. S1 Table lists the 29 cell lines with their RRIDs and tumor types (25 in the main study and 4 in the validation study).

### Generation of standardized TCM

Adherent cells were thawed, seeded into T-150 flasks with D10, grown to ~80% confluence (≥24 h), trypsinized, and $1 \times 10^6$ cells re-seeded. At ~80% confluence, media was replaced with 9 mL D10. After 24 h, TCM was collected, spun, and stored, and cells were counted. Suspension cells were seeded into T-150 flasks with R10, grown 48 h, counted, and $1 \times 10^6$ cells re-seeded. After 2–4 days, cultures were counted, then 60% of the median adherent cell number was seeded into T-150 with 9 mL D10. After 24 h, TCM was collected and cells counted.

### Primary macrophage culture

Canine peripheral blood mononuclear cells (PBMCs) from donors presented as patients to the FACC with owner's consent and under an approved Colorado State University Clinical Review Board protocol were isolated using lymphocyte separation media (Corning) by density-gradient centrifugation and plated in R20 for overnight monocyte adherence. Monocytes were differentiated for 7 days in R20 with 40 ng/mL human macrophage colony-stimulating factor (M-CSF; Peprotech Inc., #300–25), with a 50% media change every 2–3 days. For the primary TCM polarization experiment, blood from each of three donors was split into 25 wells. Macrophages were polarized for 24 h with 70% TCM/30% R10 or control medium (70% D10/30% R10), washed three times with phosphate buffered saline (PBS, Corning), incubated in R10 for 24 h, and supernatants collected. Cell-free control wells confirmed complete TCM removal. Donor characteristics (n = 3, breed, age, sex, weight, diagnosis and monocyte count) are provided in S2 Table.

### Cytokine analysis of cell supernatants

Concentrations of C-C motif chemokine ligand 2 (CCL2), vascular endothelial growth factor (VEGF), transforming growth factor-beta (TGF-β) and tumor necrosis factor-alpha (TNF-α) were measured using individual enzyme-linked

immunosorbent assay (ELISA) kits (R&D Systems Inc.: DY1774 [CCL2], DY1603 [VEGF], DY240 [TGF-β], and DY1507 [TNF-α]). In parallel, a 13-plex canine cytokine/chemokine immunology multiplex assay (MILLIPLEX®, MilliporeSigma, Burlington, MA) was performed on a Luminex® MagPix system with xPONENT® software following the manufacturer's protocol. The panel measured interferon-gamma (IFN-γ), CCL2, interleukin-8 (IL-8), IL-10, IL-2, IL-6, keratinocyte chemoattractant-like chemokine (KC-like), IL-18, TNF-α, IL-7, interferon gamma-induced protein 10 (IP-10), IL-15, and granulocyte-macrophage colony-stimulating factor (GM-CSF). Analytes below the lower limit of quantification (LLoQ) were excluded from further analysis. CCL2 exceeded the multiplex assay range and required 150-fold dilution for accurate ELISA quantification. CCL3 was measured separately by ELISA (Kingfisher: DIY1367D) in a subsequent analysis (see CCL3 assays). Data were analyzed using GraphPad Prism v10.2 (RRID: SCR_002798) and R version 4.5.1 (RRID: SCR_001905) [16,17].

## Normalization of results

Absolute values from ELISAs and the multiplex panel were normalized to best make comparisons across donors and analytes. Because the data were non-normally distributed, a modified z-score was calculated for each analyte as 0.6745 x (data point – dataset median) / median absolute deviation [18]. Due to the small number of biological replicates, no values were excluded as outliers; all data points were retained to capture the full range of biological variability. Modified z-scores were used to rank cancer cell lines, with higher values indicating greater macrophage stimulation and lower values indicating less. Scores were then correlated across analytes and with previously published mutational and phosphorylation data [19].

## RNA-sequencing and analysis

Bulk-RNA sequencing (RNA-seq) of the canine cell lines used in this study was previously performed by investigators at the FACC [20]. Spearman's rank correlations were computed between each gene's expression (transcripts per million [TPM] and removed unwanted variation using control genes [RUVg]-normalized) and cytokine modified z-scores, with $p$-values adjusted using the FDRtool R package. Differential expression between "high" and "low" macrophage stimulators (top/bottom modified z-score quartiles) was analyzed in DESeq2 using the filtered raw count matrix (excluding genes with ≤ 3 counts or detected in < 50% of samples), with internal size-factor normalization applied and $\log_2$-fold-change ($\log_2$FC) shrinkage performed using the ashr method. Genes with $|\log_2$FC$| \geq 1$ and BH-adjusted $p \leq 0.05$ were considered significant. Pre-ranked gene set enrichment analysis (GSEA) was run against Molecular Signatures Database (MSigDB) Hallmark (H), Curated (C2), Oncogenic (C6), and Immunologic (C7) gene sets using the R packages msigdbr and clusterProfiler, yielding a normalized enrichment score (NES). BH-adjusted $p \leq 0.25$ was considered significant. KEGG pathway analysis was not included as MSigDB's C2 collection incorporates KEGG-derived gene sets alongside other curated sources [21]. *CCL3* transcript counts were also extracted from public datasets (NCBI BioProject PRJDB11462 [four tumors], PRJEB36828 [seven tumors], PRJDB17594 [12 cell lines]).

## Exosome depletion and validation study

Seven top VEGF-stimulating cell lines were selected. Aliquots of their TCM were thawed and divided into whole TCM, exosome-depleted TCM (Amicon Ultra centrifugal filters; Millipore Sigma #UFC210024), and exosome-only fractions. Exosome concentrations were quantified by tunable resistive pulse sensing (qNano Gold, Izon; nanopore 150, ~47 nm stretch, calibrated with #NP100 and #CPC200 particles). Macrophages from three canine donors were differentiated for 7 days, treated with each condition (7 lines × 3 fractions), washed, and supernatants assayed for VEGF. For exosome-cargo analysis, exosome fractions were lysed in 5X RIPA buffer (volume matched to the exosome-depleted fraction) before VEGF measurement [22]. To replicate our findings, fresh TCM was generated from two high-VEGF lines (Nike, Parks),

 

two low-VEGF lines (CLL-1390, 1771), and four additional lines selected by *MVB12A* expression. Macrophages from three new donors were treated with these eight TCM, and VEGF in supernatants was measured by ELISA as above.

## CCL3 knockdown

Small interfering RNAs (siRNAs) were designed against canine CCL3 (GenBank: AB164618.1) in conjunction with a non-targeting control (NTC) RNA (targeting siRNA sequences: GAA UAU GUG GCC GAU CUG AAG CUG A; CCU UUU CUG GUG UUC CGA AAG AUA AUA CCG; CUU UUC UGG UUU CGA AGA UAA UAC CGG; and NTC: MISSION® siRNA Universal Negative Control #1, SIC001, Sigma-Aldrich). siRNA-mediated knockdown (KD) in DH82 cells was performed by seeding 50,000 cells per well in a 6-well plate in antibiotic-free medium 24 h before transfection. Gene-specific or NTC siRNAs were added at various concentrations to assess KD efficiency. Ultimately, 100 nM final concentration siRNA was complexed with HiPerfect transfection reagent (Qiagen, #301704) in Opti-MEM reduced serum media (Thermo Fisher, # 31985088) and added to the cultures according to the manufacturer's protocol. Cells were incubated for 24 h, after which the medium was replaced with D10. KD efficiency in KD, NTC and wild-type (WT) cells was assessed at 72 h post-transfection by western blotting with an anti-CCL3 antibody (Kingfisher, # KP1365D, RRID: AB_3717327) and Fiji (RRID: SCR_002285) for densitometric quantification, and by CCL3 ELISA in supernatants from the KD and WT cells at 24, 48 and 72 h post-transfection.

## CCL3 assays

Macrophages from four new canine donors were stimulated in triplicate with recombinant canine CCL3 (Novus Biologicals, Centennial, CO, #NBP3–11065) across a concentration range for 24 h. Supernatants were assayed for TNF-α by ELISA and corrected for cell loss at higher CCL3 levels. Fresh TCM was generated from WT, KD and NTC DH82 cells. Donor macrophages were treated with each TCM in triplicate for 24 h, washed, and supernatants assayed for TNF-α by ELISA.

## Statistical analysis

This was an exploratory study, and no formal power analysis was conducted. A minimum of three independent canine donors were used for all macrophage experiments. Analyses and data visualization were performed using GraphPad Prism v10.2 (RRID: SCR_002798) and R version 4.5.1 (RRID: SCR_001905) [16,17]. Normality was assessed by the Shapiro–Wilk test, and data are reported as mean ± standard error of the mean (SEM) or median (range). Statistical significance was set at $\alpha = 0.05$ unless otherwise specified, and results are denoted as: *$p < 0.05$, **$p < 0.01$, ***$p < 0.001$, ns = not significant. Intraclass correlation coefficients (ICCs) were calculated using linear mixed-effects models with treatment as a fixed effect and donor as a random effect, representing the proportion of variance due to donor identity after accounting for treatment. *P*-values from likelihood ratio tests for the seven analytes were adjusted for multiple comparisons using the Benjamini-Hochberg (BH) false discovery rate (FDR) method. Cell line and tumor type overrepresentation in top/bottom quartiles was tested using permutation analysis with 20,000 permutations (cell line) or 5,000 permutations (tumor type, 4- and 5-category grouping), a fixed random seed, and BH adjustment. Pairwise cytokine correlations and correlations between gene expression and cytokine z-scores were assessed by Spearman's rank correlation with BH adjustment. Nonparametric group comparisons were performed using the Wilcoxon rank-sum test for binary categories (e.g., mutation present vs. absent) and the Kruskal–Wallis test for multi-level factors (e.g., pAKT status). Linear mixed-effects models were applied to dose-response and exosome-fractionation experiments, specifying treatment condition as a fixed effect and donor as a random intercept. Model assumptions were verified by inspection of residual vs. fitted value plots for homoscedasticity and Q–Q plots for normality. Two-group comparisons of high- versus low-*MVB12A* expressors were analyzed using Welch's or Student's *t*-tests as appropria*t*e. One-sample *t*-tests were used for comparisons normalized *t*o a reference value (e.g., normalized to WT = 1). Multiple comparisons were corrected using the BH method across analyses.

## Results

### Canine cancer cell lines vary in macrophage polarization capacity

Using three independent canine donors and TCM from 25 cell lines, macrophage-secreted cytokines and chemokines were quantified after TCM treatment. The mean tumor cell count at TCM harvest was 11.04 × 10^6 (range 4.6–20.04 × 10^6), with no significant differences in tumor cell count across tumor types (S1 Fig). Of 15 analytes measured, seven were consistently detectable: CCL2, IL-8, IL-10, KC-like, TNF-α, TGF-β, and VEGF. The remaining eight (IL-15, IFN-γ, IL-2, IL-6, IL-7, IP-10, IL-18, and GM-CSF) had between 43–100% of data points below the LLoQ and were excluded from further analysis. Significant inter-donor variability was observed for most cytokines. CCL2 and VEGF showed the greatest variability, consistent with wide dynamic ranges and high ICCs. KC-like was marginally insignificant and showed a relatively low ICC, indicating a more consistent response across donors. Descriptive statistics, ICCs, and associated $p$-values for the seven analytes are summarized in **Table 1**. Cell lines showed marked heterogeneity in macrophage polarization capacity. Modified z-scores are shown in **Fig 1** and raw donor values in S2 Fig. Neither cell number at TCM harvest nor tumor type significantly affected analyte levels (linear mixed-effects models; all adj. $p > 0.05$), indicating results were not driven by cell density or tumor origin (S3 Table **A and B**).

Two cytokine pairs were significantly correlated after FDR adjustment, with the strongest association between IL-10 and TNF-α (ρ = 0.965, adj. $p < 0.0001$). IL-8 and KC-like also showed significant correlation (ρ = 0.656, adj. $p = 0.004$) (**Fig 2**). Therefore, TCM that induced high IL-10 secretion also tended to induce high TNF-α secretion, and TCM that induced high IL-8 secretion also tended to induce high KC-like secretion. The remaining pairwise correlations did not reach significance after correction. All pairwise comparisons from this analysis are provided in S4 Table. No cell lines or tumor types were significantly overrepresented in the top or bottom quartiles across analytes by permutation testing. Cell line 1771 (B-cell lymphoma) showed a trend toward overrepresentation in the bottom quartiles (adj. $p = 0.064$).

Whole-exome sequencing of >30 canine cell lines was previously performed at the FACC, categorizing driver mutations and phosphorylation status of protein kinase B (pAKT) and extracellular signal-regulated kinase (pERK) as active, constitutive, or low in 23 lines used here [19]. We next explored correlations between these data and our results. Three associations showed uncorrected $p < 0.05$ but did not survive FDR correction. TCM from cell lines with ≥ 2 driver mutations (n = 8) stimulated higher TGF-β compared to lines with < 2 driver mutations (n = 15) (Wilcoxon W = 20, $p = 0.011$, adj. $p = 0.075$). Conversely, lines with < 2 driver mutations stimulated higher IL-10 (W = 91, $p = 0.049$, adj. $p = 0.133$). TCM from lines with active pAKT (n = 9) stimulated higher CCL2 compared to constitutive pAKT (n = 11) (pairwise Wilcoxon $p = 0.023$, adj.

**Table 1. Analytes included for further analysis.**

| Analyte | Median | Range | ICC | $\chi^2$ | Adj. $p$-value |
|---|---|---|---|---|---|
| VEGF | 0.8998 | 4.2406 | 0.614 | 60.28 | <0.001 |
| TNF-α | 0.4402 | 16.6646 | 0.182 | 9.45 | 0.003 |
| IL-10 | 0.4153 | 40.2392 | 0.115 | 4.81 | 0.033 |
| CCL2 | 0.141 | 0.9815 | 0.754 | 92.13 | <0.001 |
| IL-8 | 0.1068 | 2.5799 | 0.209 | 11.5 | 0.001 |
| TGF-β | 0.0078 | 1.7551 | 0.628 | 62.9 | <0.001 |
| KC-like | −0.0092 | 3.8951 | 0.096 | 3.68 | 0.055 |

Data are expressed as modified z-scores. A modified z-score of +0.6745 indicates the measurement lies one median absolute deviation (MAD) above the median and of −0.6745 indicates one MAD below. Intraclass correlation coefficients (ICCs) represent the fraction of residual variance after accounting for treatment effect attributable to donor identity, and the corresponding adjusted $p$-value from a likelihood-ratio test ($\chi^2$, df = 1) comparing models with and without donor as a random effect.

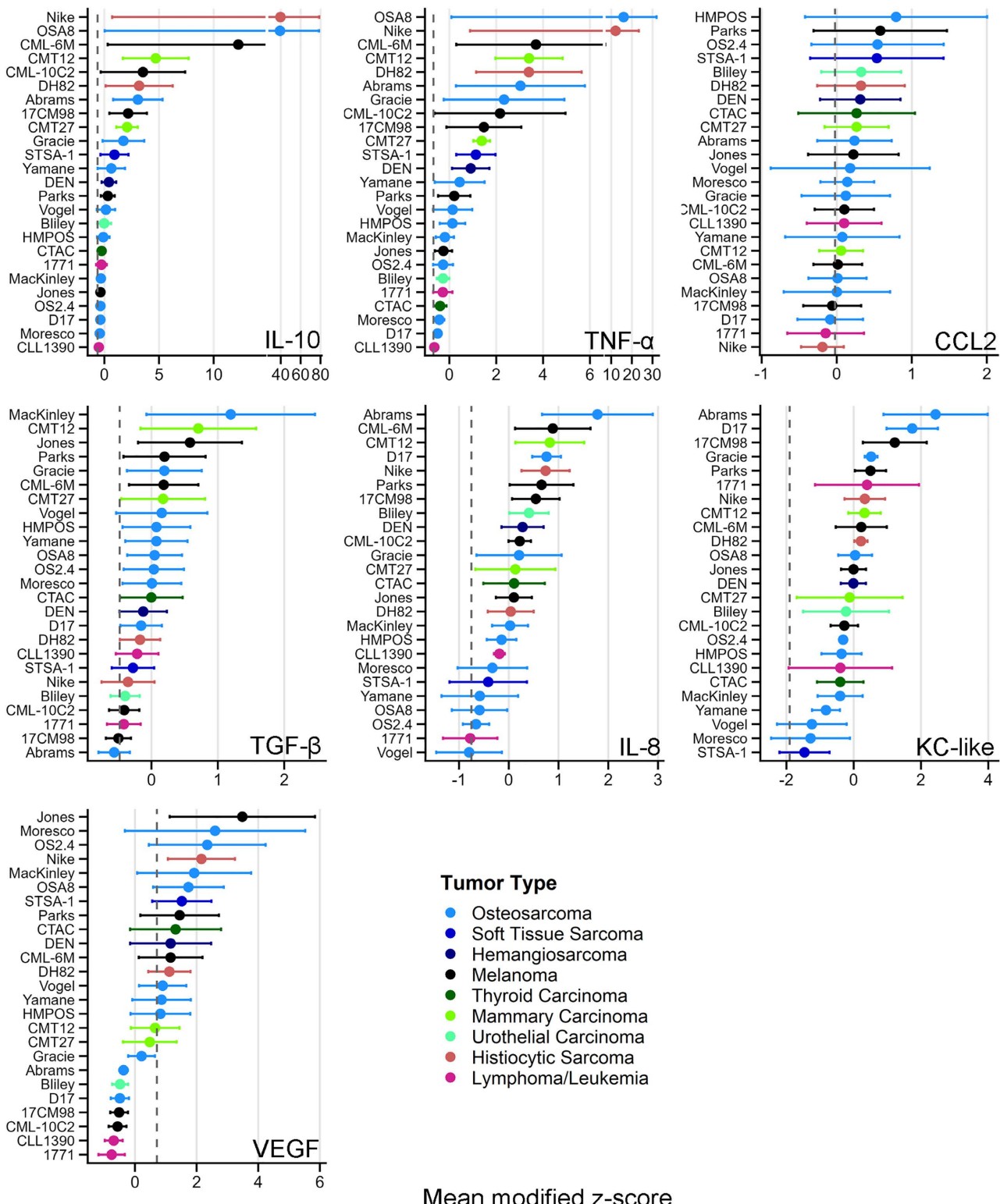

**Fig 1. Modified z-scores of each analyte.** Mean modified z-score for each cytokine secreted by donor macrophages, ranked in descending order, color-coded by tumor type of the cell line. Vertical dotted line represents the mean modified z-score of control macrophages from the same 3 donors. Data are mean ± standard error of the mean (SEM); n = 3.

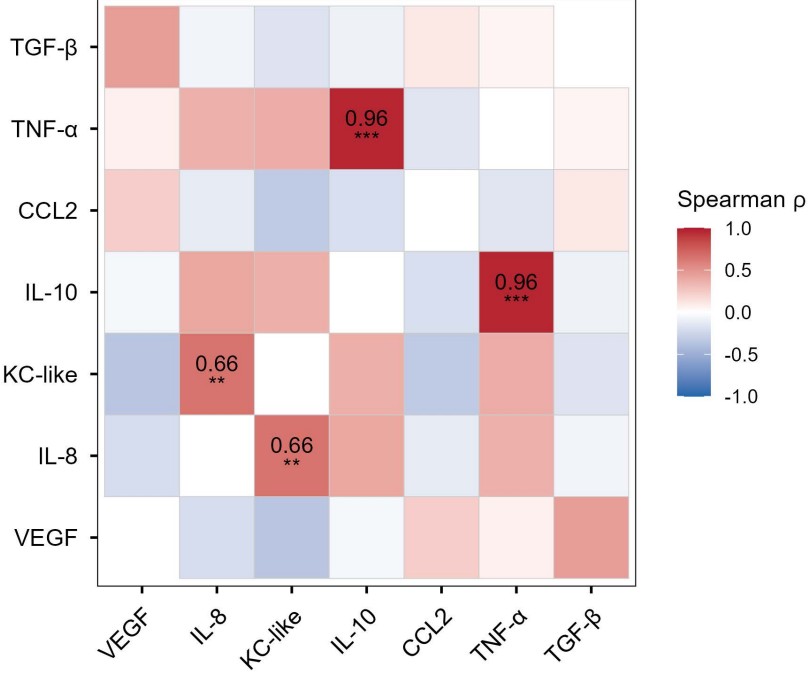

**Fig 2. Spearman correlation heatmap of cytokine secretion.** Each tile is colored by the Spearman ρ value (red = positive correlation, blue = negative correlation; scale bar at right). Statistically significant correlations after false discovery rate correction for multiple comparisons are annotated.

$p = 0.068$; overall Kruskal-Wallis H = 7.03, df = 2, $p = 0.030$, adj. $p = 0.209$). These exploratory associations could be investigated further for biological significance.

### High-polarizing cell lines share distinct transcriptomic signatures

Investigators at the FACC had previously performed RNA-seq on 23 of the 25 cell lines used in this study [20]. We next correlated cytokine modified z-scores with this RNA-seq data. For VEGF, two genes showed strong associations. Multivesicular body subunit 12A (*MVB12A* [gene ID 476674]; ρ = 0.817, $p < 0.001$, q = 0.039) was upregulated in high VEGF-stimulating cell lines and encodes a core endosomal sorting complexes required for transport (ESCRT)-I subunit regulating exosome biogenesis [23]. An unannotated Ensembl ID (ENSCAFG00000024217; ρ = 0.813, $p < 0.001$, q = 0.039) was also upregulated in high VEGF-stimulating cell lines. It was identified by ortholog analysis as trafficking protein particle complex subunit 5 (*TRAPPC5* [gene ID 484997]), part of the TRAPP complex involved in vesicular trafficking [24,25]. These results suggest a mechanistic link between tumor cell vesicle/exosome pathways and macrophage VEGF release (**Fig 3**, linear regression statistics in S5 Table).

Pairwise differential gene analysis was performed by defining "high" and "low" stimulator groups (n = 6 cell lines each) for each analyte based on the top and bottom quartiles of modified z-scores. The two groups showed significantly different cytokine responses for all seven analytes (all $p < 0.01$, S6 Table). Comparing top vs. bottom quartile cell lines, there were 210 DEGs across all cytokines comprising 188 unique genes, indicating some genes were common across multiple cytokines (**Fig 4**). Positive log₂FC indicates higher gene expression in cell lines stimulating greater macrophage secretion, while negative log₂FC indicates higher expression in low-stimulator cell lines. In high-stimulator cell lines, upregulated DEGs included macrophage activation/recruitment (*CSF1R* [gene ID 489188], *CCL7* [491148], *STAB1* [100856645], *ADAM8* [91699], *CXCL14* [610078], *IL12A* [403977]), epithelial-to-mesenchymal transition (EMT) (*SNAI1* [485924],

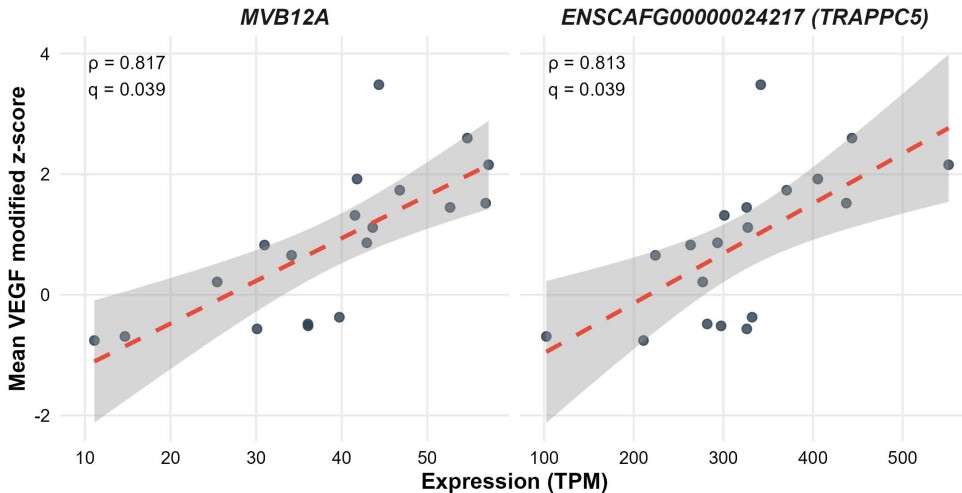

**Fig 3. Spearman's correlation between gene expression and VEGF response.** Scatter-plots showing the relationship between TPM-normalized RNA-seq expression of *MVB12A* (left panel) and *TRAPPC5* (right panel) versus mean VEGF modified Z-scores (n = 23 cell lines). Dashed red lines depict the linear regression trend with 95% confidence bands shaded in grey; annotations show Spearman ρ and q-values.

*SEMA7A* [487644]), extracellular matrix remodeling (*COL6A3* [403582]), and metabolic reprogramming (*GFPT2* [474653], *SMPDL3A* [476279]). Downregulated DEGs included immune surveillance markers (*LAPTM5* [487324], *CTSS* [403400]) and cell adhesion genes (*SEPTIN1* [489900], *ACTA2* [477587]).

We then performed GSEA with the gene lists from the pairwise analysis based on pre-ranked $\log_2$FC values using MSigDB's Hallmark, C2, C6, and C7 collections. Across all seven cytokines, 1,935 gene sets were enriched (range: 2–822 per cytokine), with the majority from immunologic signatures (C7, n = 1,092) and curated gene sets (C2, n = 704). The high-stimulator groups showed enrichment for relevant pathways such as EMT (e.g., HALLMARK_EPITHELIAL_MES-ENCHYMAL_TRANSITION in CCL2: NES = 1.78, q = 0.003), immune suppression (CCL2, TGF-β, VEGF), MEK signaling (IL-8), macrophage response/activation (CCL2, IL-10, TNF-α), extracellular matrix remodeling (TNF-α, VEGF) and suppression of IL-2 signaling (TNF-α) (**Fig 5**).

## Exosome biogenesis gene *MVB12A* drives VEGF stimulation

We validated our RNA-seq correlation findings suggesting that *MVB12A* expression in cancer cell lines is associated with VEGF stimulation from primary macrophages. *MVB12A* encodes a subunit of ESCRT-I involved in exosomal cargo sorting, leading us to test whether exosomes mediate this response [23]. As described in Methods, TCM was fractionated into whole, exosome-depleted and exosome-only conditions using 7 top VEGF-stimulating cell lines tested across 3 additional donor macrophage samples (n = 63 observations). Macrophages treated with the exosome-only fraction secreted significantly more VEGF than those treated with whole or exosome-depleted TCM (**Fig 6A**). Post-hoc comparisons showed the exosome-only fraction increased VEGF by ~1,000 pg/mL compared to both whole and exosome-depleted TCM (both p < 0.0001), while whole and exosome-depleted conditions did not differ (p = 0.879). Individual cell line analyses confirmed that exosome-only fractions significantly increased VEGF secretion in 6 of 7 lines (all except OS2.4; all pairwise comparisons are reported in S7 Table). VEGF levels were then measured in TCM from these 7 top VEGF-stimulating lines. To test whether TCM VEGF content directly drives macrophage VEGF secretion, we correlated TCM VEGF levels with macrophage VEGF responses from the same 6 donor macrophages already used in this study (n = 42 total observations).

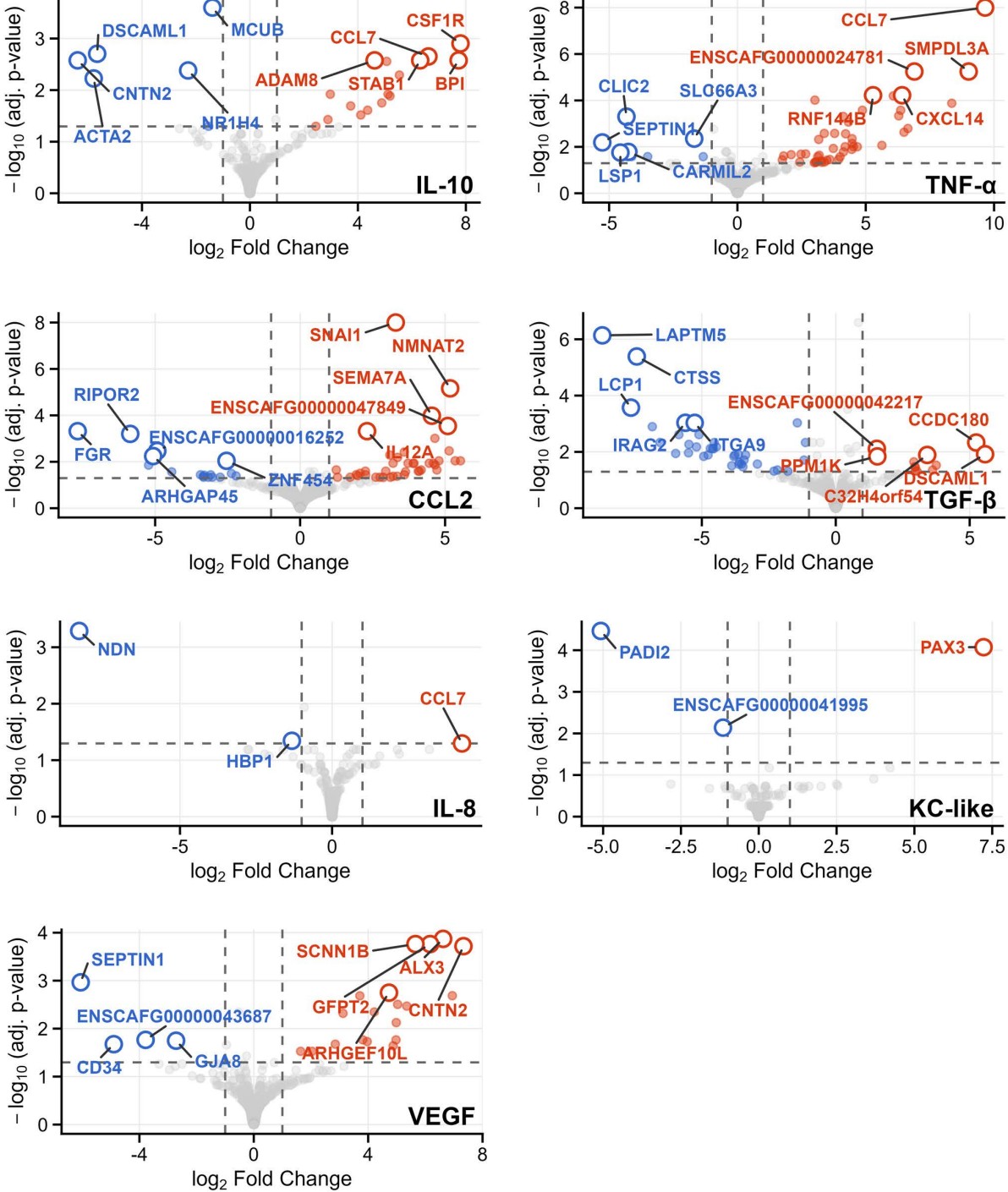

**Fig 4. Volcano plots of differentially expressed genes.** Volcano plots comparing high- vs. low-stimulator cell lines (top/bottom quartiles, n = 6 per group for each cytokine). Dashed lines indicate significance thresholds (|shrunken log$_2$FC| ≥ 1 and FDR ≤ 0.05). Significant genes are colored red (up) or blue (down). Up to the top 5 genes are annotated.

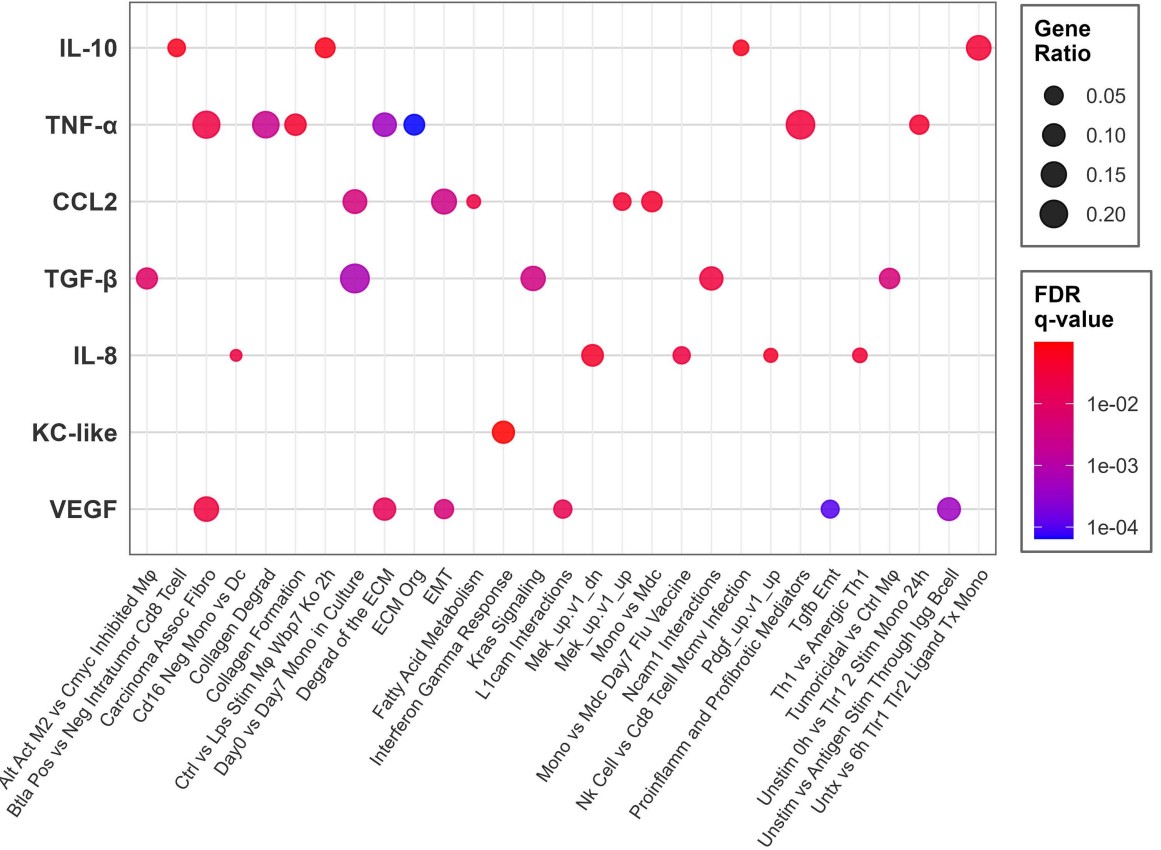

**Fig 5. Select enriched pathways across cytokines.** Gene set enrichment analysis (GSEA) pathways enriched in high- vs. low-stimulator cell lines. Dot color indicates FDR *q*-value (blue = more significant); dot size indicates gene ratio. Thresholds of FDR *q* ≤ 0.25 and |normalized enrichment score (NES)| ≥ 1.0 are shown. Pathway labels have been edited for clarity.

TCM VEGF levels were not significantly correlated with macrophage VEGF induction (mixed-effects model with donor as random intercept: β = 103.6 pg/mL per SD increase in TCM VEGF, SE = 120.8, 95% CI [−133.2, 340.5], t(35) = 0.857, *p* = 0.397), indicating that macrophage VEGF secretion is not a simple dose-response to TCM VEGF content.

To further validate this finding, we generated fresh TCM from eight cell lines: four from the original experiment (Parks, Nike, 1771, and CLL-1390) and four newly selected (CIN, SB, Angus, and Tyler1) based on *MVB12A* expression (**Fig 6B**). Freshly made TCM from these 8 lines was tested on 3 new donor macrophages. Seven of the eight lines stimulated macrophages as predicted, with high *MVB12A* associated with high VEGF. When all eight lines were analyzed, the numerical difference in VEGF stimulation between high- and low-*MVB12A* groups did not reach statistical significance (high: 507 ± 132 pg/mL, n = 4; low: 255 ± 54.8 pg/mL, n = 4; Welch's t-test, t = 1.77, df = 4.00, *p* = 0.152). CIN, a hemangiosarcoma (HSA) line, showed unexpectedly low VEGF stimulation despite high *MVB12A* expression, potentially representing a tumor type-specific deviation from the general pattern. When CIN was excluded as a potential biological outlier, high-*MVB12A* cell lines induced significantly greater VEGF secretion (630 ± 65.8 pg/mL, n = 3 vs. 255 ± 54.8 pg/mL, n = 4; t = 4.39, df = 4.34, *p* = 0.0098, 95% CI [145.2, 606.3]; **Fig 6C**). This significance persisted when both HSA lines (CIN and SB) were excluded (630 ± 65.8 pg/mL vs. 293 ± 55.8 pg/mL; t = 3.91, df = 3.90, *p* = 0.018, 95% CI [95.7, 579.9]), suggesting the *MVB12A*-VEGF relationship may differ in HSA.

To test whether exosomes carried VEGF or stimulated VEGF indirectly, TCM aliquots from the same eight lines were depleted of exosomes, producing exosome-depleted TCM and concentrated exosomes (resuspended to equal volume).

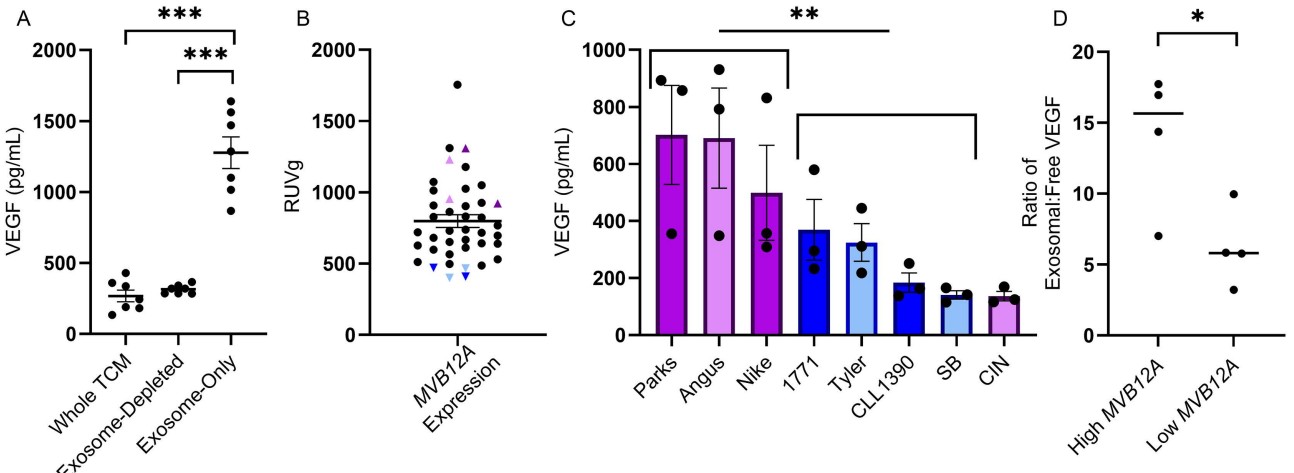

**Fig 6. Validation of *MVB12A*'s role in exosome-associated VEGF stimulation. (A)** VEGF secretion by donor macrophages (n = 3) treated with whole TCM, exosome-depleted TCM, or exosome-only fractions from 7 high-VEGF stimulating cell lines (n = 21 per condition). Linear mixed-effects model revealed a significant main effect of treatment condition (F(2, 58) = 69.46, $p < 0.0001$). Post-hoc Tukey comparisons: whole TCM vs. exosome-depleted ($p = 0.879$); whole TCM vs. exosome-only ($p < 0.0001$); exosome-depleted vs. exosome-only ($p < 0.0001$). **(B)** *MVB12A* expression (RUVg-normalized) across cell lines. For (B) and **(C)**, purple = high expressors, blue = low expressors, darker shades = discovery set, lighter shades = validation set. **(C)** VEGF secretion by donor macrophages (n = 3) treated with TCM from high- vs. low-*MVB12A* cell lines. Analysis excluding CIN (n = 7 cell lines total: 3 high, 4 low): High-*MVB12A* lines induced significantly greater VEGF (mean ± SEM: 630 ± 65.8 pg/mL) than low-*MVB12A* lines (255 ± 54.8 pg/mL; Welch's t-test, t = 4.39, df = 4.34, $p = 0.0098$, 95% CI [145.2, 606.3]). **(D)** Ratio of exosomal to free VEGF in TCM from high- vs. low-*MVB12A* expressing cell lines (n = 4 per group). High-*MVB12A* lines showed significantly greater enrichment of VEGF in exosomes compared to free VEGF (mean ± SEM: 14.01 ± 2.44 vs. 6.20 ± 1.39; Student's t-test: t = 2.78, df = 6, $p = 0.032$, 95% CI [0.94, 14.69]). Bars show mean ± SEM with individual data points overlaid.

All *MVB12A*-high lines, including CIN, had higher VEGF in the lysed exosome fraction compared to the exosome-depleted fraction, consistent with exosome-associated cargo. The ratio of exosomal to free VEGF was significantly higher in high-*MVB12A* lines (n = 4, including CIN) compared to low-*MVB12A* lines (n = 4) ($p = 0.032$; **Fig 6D**). These findings indicate that *MVB12A* expression predicts VEGF packaging into exosomes across all cell lines tested, including CIN. However, while CIN efficiently packaged VEGF into exosomes, it failed to stimulate macrophage VEGF secretion (**Fig 6C**), suggesting the functional difference in CIN occurs downstream of exosomal cargo loading, potentially at the level of exosome-macrophage interaction or uptake.

Finally, in Parks and Angus (two high-*MVB12A* lines), VEGF was compared between the concentrated exosome fraction and exosome-depleted TCM when exosomes were lysed versus unlysed. In unlysed samples, the VEGF ratio decreased substantially (Parks: 17.7-fold to 5.8-fold; Angus: 7.0-fold to 1.3-fold), suggesting much of the VEGF is packaged inside exosomes rather than surface-associated or co-isolated with them, and that lysis is required to detect the full VEGF cargo.

## CCL3 mediates TNF-α induction in histiocytic sarcoma

Preliminary inspection of unfiltered TNF-α differential expression results revealed that *CCL3* was significantly upregulated in high TNF-α-stimulating cell lines ($\log_2 FC = 24.92$, SE = 2.97, 95% CI [19.1, 30.7], Wald statistic = 8.40, adj. $p = 5.93 \times 10^{-14}$, base mean = 398.2). However, this gene was expressed in only 2 of 23 cell lines, including DH82 (10.2 TPM, TNF-α mod z = 3.39) and Nike (114.5 TPM, TNF-α mod z = 12.13), with zero detectable counts in all other lines (*CCL3* was therefore excluded from final analysis due to low prevalence). Both DH82 and Nike were in the top quartile for TNF-α stimulation and both of histiocytic sarcoma (HS) origin. Publicly available RNA-seq datasets showed that *CCL3* is broadly expressed in canine HS tumors and cell lines (NCBI BioProject PRJDB11462 [four tumors], PRJEB36828 [seven

tumors], PRJDB17594 [12 cell lines]). This suggested *CCL3* expression may be restricted to tumors of hematopoietic lineage and is commonly expressed in HS in dogs.

We next measured CCL3 protein levels in TCM from all 23 cell lines with RNA-seq data available by ELISA. As expected, DH82 and Nike secreted the highest amounts of CCL3 (597 and 675,942 pg/mL, respectively), with Nike producing >1000-fold more than DH82. Four additional lines had low but detectable CCL3 (range: 1.3–3.98 pg/mL, mean = 2.69 pg/mL) not captured at the transcript level in RNA-seq, likely due to low expression below the detection threshold. The remaining 17 lines had undetectable CCL3 (< 1.3 pg/mL) (**Fig 7A**). We confirmed the correlation between *CCL3* and TNF-α with recombinant canine CCL3, which induced a dose-dependent increase in TNF-α secretion from macrophages (linear mixed-effects model: *p* = 0.0002; 210% increase from 0 to 5 ng/mL). Results were adjusted for relative cell

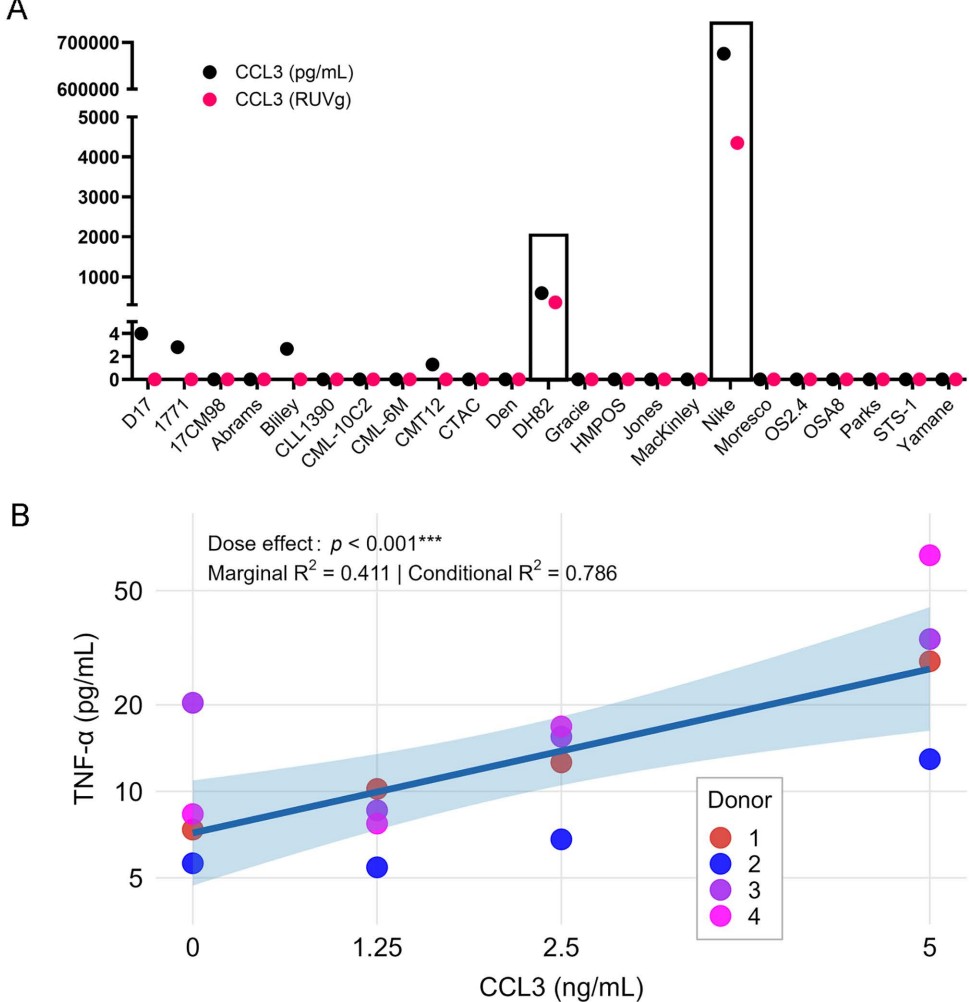

**Fig 7. *CCL3*-induced TNF-α secretion by macrophages. (A)** CCL3 protein levels in TCM measured by ELISA (black) and RUVg-normalized counts (magenta) across 23 cell lines. DH82 and Nike (both histiocytic sarcoma) showed the highest levels. Four lines had low but detectable levels (1.3-3.98 pg/mL) not detected by RNA-seq, while 17 lines were undetectable (< 1.3 pg/mL). **(B)** Dose-dependent TNF-α secretion by macrophages in response to recombinant CCL3 (n = 4 donors, 4 doses each). Linear mixed-effects model with inverse transformation (1/TNF-α) revealed a significant dose effect (β = −0.0183 on inverse scale, F(1, 11) = 28.85, *p* = 0.0002, marginal $R^2$ = 0.411, conditional $R^2$ = 0.786. Predicted means: 7.42 pg/mL at 0 ng/mL, 23.02 pg/mL at 5 ng/mL). Values adjusted for cell counts.

counts, as more macrophages were detached at higher CCL3 concentrations (**Fig 7B**, S8 Table). *CCL3* KD was assessed in DH82 cells by western blot, showing 45% of WT levels (61% of NTC), and by ELISA, showing time-dependent reduction in secreted CCL3 (70% of WT at 24h, 55% at 48-72h). To explore the functional effect of *CCL3* KD on macrophage TNF-α secretion, TCM from WT, NTC, and KD-DH82 cells was applied to macrophages from three additional canine donors. TNF-α secretion varied across donors: relative to WT, macrophages exposed to KD-DH82 TCM produced 34%, 15%, and 0.2% as much TNF-α, respectively. When normalized to the NTC condition, two donors showed decreased TNF-α secretion (to 80% and 11% of NTC), whereas one donor showed a higher response (380% of NTC). Given the small sample size and donor variability, the results are reported primarily descriptively; exploratory inferential statistics are provided in S3 Fig.

## Discussion

This study tested two main hypotheses: first, that canine tumor cell lines differ in their ability to polarize macrophages independent of tumor type, and second, that the most potent polarizers have relevant gene expression profiles. We undertook this work because the TME is a major barrier to therapy and shows significant variability even within tumor types. TME-based biomarkers could therefore help stratify patients for immunotherapies. Here, we assessed the ability of cancer cell lines to stimulate immunomodulatory cytokines in primary macrophages as a simplified proxy for *in vivo* cancer cell-macrophage crosstalk and correlated these responses with cancer cell line transcriptomes. We focused on macrophage-secreted cytokines as markers of an "active" TME rather than strict M1/M2 indicators. VEGF, TNF-α, IL-10, TGF-β, IL-8, CCL2, and KC-like showed significant variability between cell lines and are major players in the TME [26–28]. VEGF drives tumor angiogenesis and tumor cell survival, TGF-β promotes EMT, matrix remodeling, and immune cell exclusion, and CCL2 recruits immunosuppressive monocytes, myeloid-derived suppressor cells, and T-regulatory cells [28–30]. IL-8 and KC-like were moderately correlated and together recruit immunosuppressive myeloid subsets [26,31]. IL-10 and TNF-α were strongly correlated, consistent with feedback balancing pro- and anti-inflammatory signaling [32]. These cytokine patterns help illustrate the complexity of how tumors maintain their TME. While we concentrated on TCM that drove high secretion, several lines actively repressed cytokines relative to unstimulated (M0) macrophages. This is most notable with VEGF and CCL2 and warrants further investigation. DEG and GSEA analysis revealed cytokine-specific enrichment patterns, with CCL2/VEGF linked with EMT and immune suppression, TNF-α with matrix remodeling, and IL-8 with MEK signaling. Several cytokines also shared upregulation of macrophage activation and regulation genes. Further work is needed to define mechanistic links between these cytokines and genomic patterns.

Among our findings, the identification of *MVB12A* as a DEG strongly associated with VEGF induction from macrophages represents a particularly novel contribution. In people, *MVB12A* is a component of the ESCRT-I complex required for exosome biogenesis, suggesting it may function upstream of macrophage polarization by regulating the quantity or cargo of tumor-derived exosomes [23]. Exosome-mediated macrophage reprogramming has emerged as a critical mechanism in human cancers, including breast, gastric, pancreatic and colorectal carcinomas [33,34]. In these malignancies, exosomes have been shown to directly promote angiogenesis via the VEGF/VEGFR pathway as well as reprogram macrophages towards pro-tumoral phenotypes through multiple mechanisms [33,34]. However, this axis has not previously been characterized in canine tumors to the authors' knowledge. To test whether exosomes mediated VEGF induction in our system, we performed an exosome-enrichment assay and found that macrophages incubated with the purified exosome fraction secreted significantly more VEGF than those treated with whole or exosome-depleted TCM. Lysis of the exosome fraction yielded high VEGF concentrations, most consistent with direct carriage of VEGF by the exosomes, though co-isolation of contaminating proteins cannot be excluded. These findings suggest that canine tumor cells may similarly exploit exosome-dependent pathways to shape TAM phenotypes and promote angiogenesis. However, whether canine TAMs respond to exosomal cargo through the same pathways as human macrophages remains unknown. Further study with more rigorous exosome-isolation methods and larger cohorts is warranted.

   

We showed that seven of eight *MVB12A*-expressing cell lines stimulated VEGF release as expected. The exception was the HSA line CIN, which expressed high *MVB12A* but stimulated low VEGF. All *MVB12A*-high lines, including CIN, showed VEGF enrichment in exosomes compared to free VEGF, suggesting that CIN-derived exosomes may interact differently with macrophages. Another HSA line, SB, was also a low VEGF stimulator with low *MVB12A*, possibly reflecting a tumor type–specific adaptation and warranting caution in characterizing *MVB12A* as a universal predictor of macrophage VEGF-induction. Despite high VEGF secretion, canine HSA cells are insensitive to VEGF-driven proliferation, so they may gain little benefit from macrophage-derived VEGF [35]. Similarly, human angiosarcoma, a comparable malignancy, harbors mutations causing ligand-independent VEGF receptor signaling, which may reduce reliance on secreted VEGF [36]. HSA may therefore represent a tumor type in which the *MVB12A*-VEGF axis has diminished relevance, an important consideration for further study of VEGF signaling in this malignancy.

Although our efforts largely focused on tumor type–agnostic findings, we identified *CCL3* upregulation correlated with TNF-α secretion from macrophages only in HS cell lines. *CCL3*, also known as macrophage inflammatory protein 1α, recruits myeloid cells and facilitates T-cell responses in the TME [37]. HS is a rare malignancy with poor prognosis in both humans and dogs, and no accepted standard-of-care treatment [38,39]. The two HS lines with upregulated *CCL3* stimulated significantly higher TNF-α secretion, and recombinant CCL3 showed a dose-dependent relationship with TNF-α in vitro. *CCL3* KD reduced TNF-α in macrophages from two of three donors, but the variable donor response suggests more complex regulation *in vivo.* The mechanism by which CCL3 drives TNF-α was not determined. One possibility is direct binding to C-C motif chemokine receptor (CCR)1/CCR5 on macrophages, activating mitogen-activated protein kinase (MAPK) or nuclear factor kappa-light-chain-enhancer of activated B cells (NF-κB) signaling and TNF-α transcription [40]. Another is induction of an intermediary cytokine that in turn stimulates TNF-α. A recent study in mice showed that boosting CCL3 at tumor sites promoted dendritic cell recruitment, T-cell activation, and improved survival with checkpoint inhibitors [37]. While chronic TNF-α can promote immune exhaustion, regulated CCL3 appears to support immune activation rather than suppression [37,41]. Further work is needed to determine if CCL3 plays a comparable immunostimulatory role in canine HS and define how the CCL3–TNF-α axis contributes to these effects. Because *CCL3* expression appears largely restricted to hematopoietic tumors, comparative studies in canine and human HS could clarify whether the CCL3–TNF-α axis represents a lineage-specific pathway or a more general therapeutic target.

The main limitation of this study is that the *in vitro* system of TCM and monocyte-derived macrophages does not capture the full complexity of the TME *in vivo*. Monocyte-derived macrophages differ from TAMs in several important ways, and TCM provides one static snapshot of tumor-secreted factors rather than the dynamic crosstalk occurring *in vivo.* Additionally, our 24-hour exposure does not reflect the sustained conditioning that shapes TAM phenotypes over months to years in developing tumors. These factors may limit the translatability of our findings to *in vivo* and clinical settings. Another limitation is that the multiplex bead-based assay and individual ELISAs had different sensitivities and dynamic ranges, so direct comparison of absolute analyte values is not recommended. We examined only the relationship between cell line transcriptomes and macrophage secretory products, without establishing direct links between proteins secreted in TCM and those released by macrophages. Protein secretion may not match mRNA levels due to post-transcriptional regulation, as shown by CCL3 protein secretion despite undetectable transcripts in some lines. Future studies integrating proteomic TCM analysis with macrophage phenotyping would help define mechanistic connections between tumor-secreted factors and macrophage polarization. Finally, the sample size of 25 cell lines and a small number of donors enabled proof-of-concept validation but was insufficient to fully capture patient variability and assess whether donor disease influenced macrophage responses. Expanded studies are needed to better characterize inter-donor variability and patient diversity.

## Conclusions

In summary, previous studies have identified multiple TAM states in human and canine tumors [3–5]. Our approach builds on this concept by tracking functional secretory outputs following TCM exposure, providing phenotypic measurements of

tumor cell signals. Measuring active cytokine release from macrophages complements static gene expression analyses. In particular, the association between *MVB12A* expression and macrophage VEGF induction identifies a potential exosome-dependent mechanism by which canine tumors may shape TAM phenotypes. This pathway has been implicated in metastasis and therapeutic resistance in human cancers but has not previously been explored in canine tumors. If validated, this axis could represent both a predictive biomarker for TME-shaping capacity and a candidate therapeutic target. However, substantial additional work, including further validation of the *MVB12A*-exosome-VEGF axis, *in vivo* studies, and ultimately evaluation in clinical cohorts would be required before this marker could be considered for clinical application. Overall, these data highlight conserved macrophage-modulating pathways across tumor types that may be exploitable as potential therapeutic targets or prognostic biomarkers in both canine and human oncology.

## Supporting information

**S1 File. Raw images. Unedited western blots for *CCL3* knockdown.** Full-membrane chemiluminescent images captured on a BioRad ChemiDoc system for CCL3 (~10 kDa, top panels) and α-tubulin loading control (~50 kDa, bottom panels). Left panels show standard chemiluminescence exposures; right panels show merged chemiluminescence and colorimetric membrane images. Lanes are labeled as follows: L, molecular weight ladder; lanes 1–5, Nike cells (not used in analysis, marked with X); lanes 6–10, DH82 cells. Treatment conditions were: 100 nM (lanes 1, 6), 50 nM (lanes 2, 7), and 10 nM (lanes 3, 8) *CCL3*-targeting siRNA; non-targeting control siRNA (lanes 4, 9); and wild-type untreated cells (lanes 5, 10). Cell lysates were collected 72 h post-transfection. Cropped regions from lanes 6–10 were used for densitometric quantification in S3 Fig.
(PDF)

**S1 Fig. Tumor cell counts by cell line tumor type.** Top: five-category grouping with osteosarcoma shown separately from other sarcomas. Bottom: four-category grouping with osteosarcoma grouped with other sarcomas. No differences were detected between groups (one-way analysis of variance: five-category $F_{(4,20)} = 1.488$, $p = 0.243$; four-category $F_{(3,21)} = 1.645$, $p = 0.209$). Data met ANOVA assumptions. Points represent individual donor measurements; box plots show median and quartiles.
(TIFF)

**S2 Fig. Raw values for each analyte, color-coded by donor.**
(TIF)

**S3 Fig. *CCL3* knockdown validation and effect on macrophage TNF-α secretion.** (A) Densitometric quantification of western blot bands from DH82 cell lysates collected 72 h after transfection with increasing concentrations of CCL3-targeted siRNA (10–100 nM), non-targeting control (NTC) siRNA, or untreated wild-type (WT) cells. Band intensities were normalized to WT levels. (B) Absolute CCL3 concentrations measured by ELISA in culture supernatants from WT, 25 nM, and 100 nM siRNA-treated DH82 cells at 24, 48, and 72 h post-transfection. (C) TNF-α secretion from three canine donor macrophages following 24 h exposure and 24 h washout to tumor-conditioned medium (TCM) from WT, NTC, or *CCL3*-KD (KD) DH82 cells. TNF-α values for each donor were normalized to the WT condition (set to 1). Both KD- and NTC-derived TCM induced significantly lower TNF-α secretion than WT (one-sample *t*-test vs 1, $p = 0.0106$ and $0.0159$ respectively), whereas KD did not differ significantly from NTC when analyzed by either a one-sample *t*-test normalized to NTC = 1 or by a paired two-sample *t*-test ($p > 0.05$). Color-coded by donor.
(TIF)

**S1 Table. Canine cancer cell lines used.** Asterisks denote cell lines used only in the validation study.
(DOCX)

**S2 Table. Donor characteristics for initial tumor-conditioned media generation.** MC = male castrated, MI = male intact, OSA = osteosarcoma, L = low (below reference range).
(DOCX)

**S3 Table. Effects of tumor cell count and tumor histology on cytokine secretion.** Linear mixed-effects model results with donor as a random intercept to account for repeated measures. **(A)** Cell count effects: Association between cell count (per million) and cytokine secretion. Data were analyzed using raw values or log(x + 1) transformation, selected based on residual normality diagnostics (Shapiro-Wilk test). Some cytokines showed residual non-normality despite transformation; in these cases, the better fit was chosen. Table reports the slope (cell count effect) for each cytokine. **(B)** Cell line tumor type effects: Association between cell line tumor type and log-transformed cytokine secretion using 4-category and 5-category classification systems. Table reports all coefficient estimates (intercept and tumor type contrasts) plus overall F-statistics with degrees of freedom for the main effect of histology. Neither cell count nor cell line tumor type significantly affected cytokine secretion after FDR correction.
(XLSX)

**S4 Table. Pairwise Spearman rank correlations between cytokine secretion profiles.** All possible pairwise correlations (n = 21) are shown with raw and FDR-adjusted *p*-values (Benjamini-Hochberg method).
(CSV)

**S5 Table. Linear regression statistics for Fig 3.**
(CSV)

**S6 Table. Top and bottom quartile cell lines used for differential gene expression analysis.** Cell lines were ranked by mean modified z-scores and divided into top and bottom quartiles for each cytokine (n = 6 per group for most cytokines). Statistical tests confirm significant differences in cytokine stimulation between groups. Test selection based on normality: Welch's t-test or Student's t-test for normally distributed data, Mann-Whitney U test otherwise. These quartile groups were then used for DESeq2 differential expression analysis.
(DOCX)

**S7 Table. VEGF response to TCM fractionation in individual cell lines.** All pairwise comparisons from linear mixed-effects models (donor as random intercept) assessing VEGF secretion by macrophages treated with whole TCM, exosome-depleted TCM, or exosome-only fractions. Negative estimate values indicate the first condition induced lower VEGF than the second. Estimate = difference in VEGF (pg/mL); SE = standard error; 95% CI = 95% confidence interval; df = degrees of freedom; t = t-statistic; p-value = Tukey-adjusted; Sig = significance level.
(CSV)

**S8 Table. Linear mixed-effects regression analysis of CCL3 dose effect on TNF-α secretion.** Model included donor as random effect (n = 4 donors, 4 doses per donor, 16 total observations).
(DOCX)

## Acknowledgments

The authors thank Rupa Idate for extensive work in maintaining and validating cell lines and Laurel Haines for generously sharing her expertise on exosome biology.

## Author contributions

**Conceptualization:** Rachel V. Brady, Douglas H. Thamm.

**Data curation:** Rachel V. Brady, Sunetra Das, Dawn L. Duval, Kristen B. Farrell, Eric P. Palmer.

**Formal analysis:** Rachel V. Brady, Sunetra Das, Eric P. Palmer.

**Funding acquisition:** Rachel V. Brady, Dawn L. Duval, Douglas H. Thamm.

**Investigation:** Rachel V. Brady.

**Methodology:** Rachel V. Brady, Sunetra Das, Dawn L. Duval, Kristen B. Farrell, Eric P. Palmer, Douglas H. Thamm.

**Resources:** Dawn L. Duval, Douglas H. Thamm.

**Software:** Dawn L. Duval.

**Supervision:** Sunetra Das, Dawn L. Duval, Kristen B. Farrell, Douglas H. Thamm.

**Validation:** Rachel V. Brady.

**Visualization:** Rachel V. Brady.

**Writing – original draft:** Rachel V. Brady.

**Writing – review & editing:** Sunetra Das, Dawn L. Duval, Kristen B. Farrell, Eric P. Palmer, Douglas H. Thamm.

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
