## [Decision Letter · Decision Letter 0]

30 Dec 2025

Thank you for submitting your manuscript to PLOS ONE. After careful consideration, we feel that it has merit but does not fully meet PLOS ONE’s publication criteria as it currently stands. Therefore, we invite you to submit a revised version of the manuscript that addresses the points raised during the review process.

Please address the comments from the reviewers in the revision.

We look forward to receiving your revised manuscript.

Kind regards,

Daotai Nie, Ph.D.

Academic Editor

PLOS One

Journal Requirements:

[NIH Ruth L. Kirschstein Institutional National Research Service Award Training Grant (T32OD01043) awarded through Colorado State University.

University of Colorado Anschutz Medical Campus Cancer Center Genomics Shared Resource Core Facility (Cancer Center Support Grant P30CA046934).

A gift from the Anschutz Foundation.].

6. We note that you have included the phrase “unpublished data” in your manuscript. Unfortunately, this does not meet our data sharing requirements. PLOS does not permit references to inaccessible data. We require that authors provide all relevant data within the paper, Supporting Information files, or in an acceptable, public repository. Please add a citation to support this phrase or upload the data that corresponds with these findings to a stable repository (such as Figshare or Dryad) and provide and URLs, DOIs, or accession numbers that may be used to access these data. Or, if the data are not a core part of the research being presented in your study, we ask that you remove the phrase that refers to these data.

Reviewers' comments:

Reviewer's Responses to Questions

**Comments to the Author**

1. Is the manuscript technically sound, and do the data support the conclusions?

Reviewer #1: Yes

2. Has the statistical analysis been performed appropriately and rigorously?

Reviewer #1: Yes

3. Have the authors made all data underlying the findings in their manuscript fully available?

Reviewer #1: Yes

4. Is the manuscript presented in an intelligible fashion and written in standard English?

Reviewer #1: Yes

Reviewer #1: Summary

This manuscript addresses an important and timely topic in comparative oncology by interrogating how tumor-intrinsic genomic programs shape macrophage polarization in canine cancers. The experimental approach—combining tumor-conditioned media, primary canine macrophages, transcriptomics, and functional cytokine readouts—is well executed and generates biologically meaningful insights. In particular, the identification of exosome-mediated VEGF induction linked to MVB12A is a compelling and relatively underexplored mechanism that distinguishes this work from parallel human studies.

Conceptually, the study nicely illustrates a central but often under-emphasized principle: cancer should be viewed as two interacting diseases—one of the malignant cell and one of the immune response. The data presented here strongly support this dual-disease framework and demonstrate how tumor genomic programs actively sculpt immunosuppressive macrophage phenotypes.

That said, several issues related to framing, structure, and emphasis limit the manuscript’s impact in its current form, particularly for a readership interested in canine immunotherapy and companion-animal oncology.

⸻

Major Comments

1. Introduction is overly human-centric and insufficiently focused on companion animals

While the Introduction is comprehensive, the majority of cited literature and conceptual framing is centered on human cancer biology and human immunotherapy, with canine cancer introduced relatively late and briefly. Given that this is explicitly a canine study, the Introduction should be re-structured to:

• Center companion animals earlier and more prominently

• Expand discussion of what is currently unknown in canine tumor immunology, rather than primarily summarizing human TME concepts

• Clarify how canine cancer is not merely a translational bridge, but a biologically informative system in its own right

As written, the manuscript risks giving the impression that the canine work is primarily confirmatory of human paradigms, whereas the data clearly demonstrate distinctive biology, especially regarding macrophage polarization and exosome-mediated signaling.

2. Tumor microenvironment section is disproportionately long relative to canine immunotherapy

The general TME overview (heterogeneity, metabolism, epigenetics, therapy resistance) is overly extensive, whereas the section on canine immunotherapy and TAM biology is comparatively brief. I recommend:

• Substantially shortening the generic TME discussion

• Redirecting space toward known gaps in canine immunotherapy, including the paucity of validated biomarkers and limited mechanistic understanding of immune suppression in dogs

• Explicitly positioning this study as addressing a major unmet need in canine cancer research

3. “Biomarker” is used too broadly and should be better qualified

Throughout the manuscript, the term biomarker is used somewhat loosely. While the data identify candidate correlates (e.g., MVB12A, CCL3), substantial additional work would be required before these could function as predictive or clinical biomarkers.

I suggest:

• Using terms such as candidate biomarker, putative biomarker, or biological correlate

• Explicitly acknowledging that significant additional validation is required, particularly in vivo and in clinical canine cohorts

This clarification would improve conceptual rigor and avoid over-interpretation.

4. Exosome-mediated macrophage activation is the most novel finding but is under-emphasized

The exosome findings represent the most innovative and distinguishing aspect of the study. The demonstration that tumor-derived exosomes drive macrophage VEGF production—and that this mechanism differs across tumor types—is a genuinely exciting contribution.

However, this mechanism is currently buried within the Results and Discussion. I strongly recommend:

• Elevating the exosome story more prominently in the Abstract and Discussion

• Explicitly highlighting how this mechanism may differ from what is known in human cancers

• Framing exosome-mediated macrophage reprogramming as a key feature of canine tumor immunology, not just a mechanistic aside

This would significantly enhance the manuscript’s impact and novelty.

5. Manuscript structure and figure placement impede readability

The current structure, with long figure legends interspersed throughout the Results, makes the manuscript difficult to read. The flow of the text is frequently interrupted.

I suggest:

• More clearly separating Results text from figure legends

• Using clearer subsection headers to guide the reader through the narrative

• Ensuring that conceptual advances are described in the text before directing the reader to figures

Improving structure would greatly enhance accessibility.

⸻

Minor and Technical Comments

1. Supplementary tables

Supplementary tables are frequently referenced, but their contents are not always sufficiently described in the main text. Brief summaries (e.g., variables included, number of samples, analytical purpose) would reduce the need for readers to constantly move back and forth.

2. R scripts and reproducibility

The manuscript states that analyses were conducted in R, but it is unclear:

• Which scripts were used

• Whether these scripts are available

• Where readers can access them

Please clarify whether the R code can be shared (e.g., GitHub, Zenodo) in line with open-science best practices.

3. Pathway analysis choices

The use of MSigDB is appropriate, but it would be helpful to explain why KEGG libraries were not included, particularly given their frequent use in comparative and metabolic pathway analysis. A brief justification would strengthen transparency.

4. Clarification of cell counts

The phrase “cell counts at TCM harvest” is confusing and reads awkwardly. It is unclear whether this refers to tumor cells, macrophages, or another population. Reporting PBMC or macrophage yields would be more intuitive and informative.

5. Canine-specific immunology emphasis

Several Discussion sections revert to human analogies. While comparative insights are valuable, the manuscript would benefit from more explicit statements highlighting what we still do not understand about canine immune regulation, and how this study opens those questions.

⸻

Final Recommendation

This is a strong and conceptually important study that addresses a real gap in canine immuno-oncology and offers mechanistic insights that are not easily derived from murine or human systems alone. With revisions that sharpen the canine focus, rebalance the Introduction, improve structure, and elevate the exosome findings, the manuscript would make a valuable contribution to the field.

.

Reviewer #1: **Yes:** Carsten KriegCarsten KriegCarsten KriegCarsten Krieg

You may also use PLOS’s free figure tool, NAAS, to help you prepare publication quality figures: https://journals.plos.org/plosone/s/figures#loc-tools-for-figure-preparation

---

## [Author Response · Author response to Decision Letter 1]

11 Feb 2026

Response to Reviewers

Manuscript ID: PONE-D-25-56168

Title: Distinct tumor genomic signatures underlie canine macrophage polarization

Dear Dr. Nie and Reviewer,

Thank you very much for the opportunity to revise our manuscript and for the very thoughtful and constructive feedback. We appreciate the reviewer's recognition of the study's contributions to comparative oncology and canine immuno-oncology. We have carefully addressed all comments and believe the revisions have substantially strengthened the manuscript.

Below, we provide point-by-point responses to each comment. Reviewer comments are shown in bold, and our responses follow.

Journal Requirements

Requirement 1: PLOS ONE Style Requirements

Response: We have revised the manuscript to comply with PLOS ONE style guidelines. Specific changes include: state abbreviations in affiliations spelled out in full (e.g., "Colorado" instead of "CO"); figure citations use "Fig" format; continuous line numbering implemented throughout; all abbreviations defined at first use; and Vancouver reference style verified. All figure and supporting information files have been fixed to comply with PLOS ONE conventions.

Requirement 2: Code Sharing

Please review our guidelines on code sharing and ensure that your code is shared in a way that follows best practice.

Response:

All R scripts used for statistical analysis, visualization, and RNA-sequencing analysis have been deposited in a public GitHub repository with a DOI from Zenodo.

We would like to update our Data Availability statement as follows:

"Analysis code is available on GitHub (https://github.com/rbrady783/TCM-macrophages) and archived on Zenodo (DOI: 10.5281/zenodo.18499208). Raw data are available on Dryad (DOI: 10.5061/dryad.brv15dvnx).”

(Raw data are available on Dryad. Please note that the Dryad DOI (10.5061/dryad.brv15dvnx) is reserved and will become active upon publication of the manuscript. In the meantime, the raw data can be accessed via Dryad's pre-publication review link: http://datadryad.org/share/LINK_NOT_FOR_PUBLICATION/4TdabHt6t8I-dNvhnq9Mp3wjyTFsqiWv2u_H1s9MUvE).

Requirement 3: Role of Funders

Please state what role the funders took in the study.

Response: The following statement is correct: "The funders had no role in study design, data collection and analysis, decision to publish, or preparation of the manuscript."

Requirement 4: Data Availability

We strongly recommend all authors decide on a data sharing plan before acceptance.

Response: All data underlying the findings are now available. See above for DOIs.

Please note that the Dryad DOI (10.5061/dryad.brv15dvnx) is reserved and will become active upon publication of the manuscript. In the meantime, the raw data can be accessed via Dryad's pre-publication review link: http://datadryad.org/share/LINK_NOT_FOR_PUBLICATION/4TdabHt6t8I-dNvhnq9Mp3wjyTFsqiWv2u_H1s9MUvE.

Requirement 5: Blot/Gel Images

Please provide the original uncropped and unadjusted images underlying all blot or gel results.

Response: Original uncropped western blot images for CCL3 knockdown validation are provided in S1_raw_images. These images show the full membrane for both CCL3 and the α-tubulin loading control, with lanes clearly labeled.

Requirement 6: "Unpublished Data" References

PLOS does not permit references to inaccessible data.

Response: The reference to unpublished data regarding CCL3 expression in canine histiocytic sarcoma has been revised. We now only cite publicly available RNA-seq datasets (NCBI BioProjects PRJDB11462, PRJEB36828, PRJDB17594) that support the finding that CCL3 is broadly expressed in canine HS tumors and cell lines. The personal communication reference has also been removed.

Reviewer #1 Comments

We thank Reviewer #1 for the thoughtful and expert evaluation of our work. The reviewer's comments have significantly improved the manuscript's focus and impact.

Major Comments

Major Comment 1: Introduction is overly human-centric

"While the Introduction is comprehensive, the majority of cited literature and conceptual framing is centered on human cancer biology... The Introduction should be re-structured to center companion animals earlier and more prominently."

Response: We agree that the Introduction was imbalanced. We have substantially revised it to: (1) center canine cancer biology earlier, establishing companion animal oncology as the primary focus; (2) expand discussion of knowledge gaps in canine tumor immunology, including the paucity of validated biomarkers and limited mechanistic understanding of macrophage-tumor crosstalk in dogs; (3) position canine cancer as biologically informative in its own right, not merely a translational bridge; and (4) reduce generic human TME content.

Major Comment 2: TME section disproportionately long

"The general TME overview is overly extensive, whereas the section on canine immunotherapy and TAM biology is comparatively brief."

Response: We have rebalanced the Introduction by substantially condensing the generic TME discussion (heterogeneity, metabolic reprogramming, epigenetic regulation) and expanding canine-specific content, including known gaps in canine immunotherapy development, limited availability of validated immune biomarkers in veterinary oncology, and specific challenges in translating TAM-targeted therapies to companion animals.

Major Comment 3: "Biomarker" used too broadly

"The term biomarker is used somewhat loosely... substantial additional work would be required before these could function as predictive or clinical biomarkers."

Response: We agree and have revised terminology when referring to our own findings, using 'candidate biomarker,' 'potential biomarker,' or similar qualifiers. We have added explicit acknowledgment in the Discussion that significant additional validation is required before these correlates could serve as predictive or therapeutic biomarkers. We have also slightly expanded some of our limitations to more fully explain the weaknesses of our experimental system.

Major Comment 4: Exosome findings under-emphasized

"The exosome findings represent the most innovative and distinguishing aspect of the study... I strongly recommend elevating the exosome story more prominently."

Response: We thank the reviewer for this important suggestion. We have: (1) revised the Abstract to more prominently feature the exosome findings; (2) moved the exosome Discussion section earlier and expanded it to highlight how this mechanism may differ from human systems and to frame exosome-mediated macrophage reprogramming as a key feature of canine tumor immunology; (3) added a more prominent Results subheading; and (4) highlighted the exosome finding as a primary contribution in the Conclusions.

Major Comment 5: Manuscript structure and figure placement

"The current structure, with long figure legends interspersed throughout the Results, makes the manuscript difficult to read."

Response: We appreciate this feedback. We understood PLOS ONE submission guidelines to require figure legends be included within the manuscript text rather than as separate captions attached to figures. We have placed boxes around figure legends to help visually distinguish them from the main text. We have also tried to improve descriptive subsection headers throughout Results to guide the reader through the narrative and ensure conceptual advances are described in the text before directing readers to figures. We have shortened some of our Figure legends as well.

Minor and Technical Comments

Minor Comment 1: Supplementary tables

"Supplementary tables are frequently referenced, but their contents are not always sufficiently described in the main text."

Response: We have added brief in-text descriptions for each supplementary table when first referenced, including the variables included, sample sizes, and analytical purpose.

Minor Comment 2: R scripts and reproducibility

"It is unclear whether these scripts are available and where readers can access them."

Response: All R scripts have been deposited in a public GitHub repository with an archived DOI via Zenodo. The repository includes data processing scripts, statistical analysis code, DESeq2 pipeline, GSEA scripts, and figure generation code. The repository URL and Zenodo DOI have been added to the Data Availability Statement.

Minor Comment 3: Pathway analysis choices (KEGG)

"It would be helpful to explain why KEGG libraries were not included."

Response: We have added a brief justification in the Methods: " KEGG pathway analysis was not included as MSigDB's C2 collection incorporates KEGG-derived gene sets alongside other curated sources."

Minor Comment 4: "Cell counts at TCM harvest" clarification

"The phrase 'cell counts at TCM harvest' is confusing... It is unclear whether this refers to tumor cells, macrophages, or another population."

Response: We have clarified this throughout the manuscript. The phrase now reads 'tumor cell count at TCM harvest' to unambiguously refer to cancer cell counts.

Minor Comment 5: Canine-specific immunology emphasis

"Several Discussion sections revert to human analogies... The manuscript would benefit from more explicit statements highlighting what we still do not understand about canine immune regulation."

Response: We have revised the Discussion to lead with canine-specific findings before drawing human parallels, add explicit statements about gaps in canine knowledge, and frame comparative insights as bidirectional i.e. what canine studies can teach us about human biology.

Additional Revisions

In addition to addressing reviewer comments, we made the following clarifications:

CCL3 knockdown quantification: We revised the description of CCL3 knockdown efficiency to use consistent percentage language. The original text stated “71% reduction vs NTC (55% vs WT),” which mixed reduction terminology with residual expression and contained an arithmetic error in the NTC comparison. Based on the raw densitometry values (normalized to α-tubulin loading control), this has been corrected to “45% of WT levels (61% of NTC)” to match the “% of” phrasing used elsewhere in the manuscript (e.g., ELISA results).

Excluded analytes moved to Results: The sentence describing eight analytes excluded due to measurements below the lower limit of quantification (LLoQ) has been moved from Methods (Cytokine analysis) to Results, as this was a data-driven finding rather than a pre-specified methodological decision.

Closing

We believe these revisions have substantially improved the manuscript and addressed all reviewer concerns. We thank the editor and reviewer for their time and constructive feedback.

Very Sincerely,

Rachel V. Brady, DVM, PhD

---

## [Decision Letter · Decision Letter 1]

17 Mar 2026

Distinct tumor genomic signatures underlie canine macrophage polarization

PONE-D-25-56168R1

Dear Dr. Brady,

We’re pleased to inform you that your manuscript has been judged scientifically suitable for publication and will be formally accepted for publication once it meets all outstanding technical requirements.

Kind regards,

Daotai Nie, Ph.D.

Academic Editor

PLOS One

Additional Editor Comments (optional):

Reviewers' comments:

Reviewer's Responses to Questions

**Comments to the Author**

Reviewer #1: All comments have been addressed

2. Is the manuscript technically sound, and do the data support the conclusions?

Reviewer #1: Yes

3. Has the statistical analysis been performed appropriately and rigorously?

Reviewer #1: Yes

4. Have the authors made all data underlying the findings in their manuscript fully available?

Reviewer #1: Yes

5. Is the manuscript presented in an intelligible fashion and written in standard English?

Reviewer #1: Yes

Reviewer #1: All of my points have been thoroughly addressed. Thank you for the thoughtful responses and for your continued efforts—this work is clearly important and very much needed.

.

Reviewer #1: No

---

## [Editor Report · Acceptance letter]

PONE-D-25-56168R1

PLOS One

Dear Dr. Brady,

I'm pleased to inform you that your manuscript has been deemed suitable for publication in PLOS One. Congratulations! Your manuscript is now being handed over to our production team.

Kind regards,

on behalf of

Dr. Daotai Nie

Academic Editor

PLOS One